# Giant spatial anisotropy of magnon lifetime in altermagnets

António T. Costa[1,2⋆], João C. G. Henriques[1,3] and Joaquín Fernández-Rossier[1,4]

**1** International Iberian Nanotechnology Laboratory (INL), Av. Mestre José Veiga, 4715-330 Braga, Portugal
**2** Physics Center of Minho and Porto Universities (CF-UM-UP), Universidade do Minho, Campus de Gualtar, 4710-057 Braga, Portugal
**3** Universidade de Santiago de Compostela, 15782 Santiago de Compostela, Spain
**4** On permanent leave from Departamento de Física Aplicada, Universidad de Alicante, 03690 San Vicente del Raspeig, Spain

⋆ antonio.costa@inl.int ,

## Abstract

Altermagnets are a new class of magnetic materials with zero net magnetization (like antiferromagnets) but spin-split electronic bands (like ferromagnets) over a fraction of reciprocal space. As in antiferromagnets, magnons in altermagnets come in two flavours, that either add one or remove one unit of spin to the $S = 0$ ground state. However, in altermagnets these two magnon modes are non-degenerate along some directions in reciprocal space. Here we show that the lifetime of altermagnetic magnons has a very strong dependence on both flavour and direction. Strikingly, coupling to Stoner modes leads to a complete suppression of magnon propagation along selected spatial directions. This giant anisotropy will impact electronic, spin, and energy transport properties and may be exploited in spintronic applications.

# 1  Introduction

The recent recognition of altermagnets as a new class of magnetic materials [1–3], originally predicted by Pekar and Rashba in 1964 [4], has been a very exciting development for both condensed matter and materials physics. In a static configuration, altermagnets camouflage very well as antiferromagnets; however, when you look under the hood the disguise is given away by the spin-polarized electronic bands. It is their dynamics, however, that reveal their true colors [5, 6]. To understand the dynamical properties of a magnetic system it is essential to look at its elementary spin excitations, or magnons [7].

A magnon in a ferromagnetic solid is usually associated to processes by which the total magnetization of the sample is lowered by the equivalent of a quantum of angular momentum, $\hbar$, and associated with the spin-lowering operator $S^-$. We thus say that a ferromagnetic magnon carries spin $S^z = -1$. In terms of elementary electronic processes, generating a magnon consists in promoting an electron from the majority spin band ($\uparrow$) to the minority spin band ($\downarrow$), and is associated with the operator $a_\downarrow^\dagger a_\uparrow$. By virtue of electron-electron interactions, the electron and the hole involved in this process form a bound state, whose energy depends on the net crystal momentum of the pair.

In antiferromagnets, magnons can have either $S^z = -1$ or $S^z = 1$, associated with lowering the spin of the $\uparrow$ sublattice or raising the spin of the $\downarrow$ sublattice. Due to the complete equivalence between the two spin directions, the two kinds of antiferromagnetic magnons ($S^z = \pm 1$) have identical energies [8]. On the other hand, it has been noted [2, 9] that magnons in altermagnets have unique features when compared to their antiferromagnetic counterparts. The most noticeable difference is that $S^z = -1$ and $S^z = 1$ magnons have distinct energies along certain directions in the reciprocal space, the same direction associated with the spin-split electronic bands.

In metallic magnets, magnons have finite lifetimes, due to the fact that they can decay into uncorrelated electron-hole pairs, also known as a Stoner excitations [10, 11]. The decay probability (hence the inverse of the magnon lifetime) is proportional to the spectral density associated with the Stoner excitations, which usually increases monotonically with energy for a fixed wavevector. Thus, magnon lifetimes typically decrease monotonically as the magnon energy increases [12].

It has been assumed hitherto [9] that, due to the distinct energies of $S^z = \pm 1$ magnons in altermagnets, their lifetimes would also be different, in an almost trivial manner. Other works have looked into the effects of magnon-magnon interactions on magnon lifetimes, a mechanism that is supposed to be relevant for insulating magnets. [13] Apart from that, very little attention has been paid to the lifetime of magnons in altermagnets, and most theoretical approaches employ spin-only models in their description [9, 14–16].

Here we show that Stoner damping in metallic and slightly doped altermagnets has highly non-trivial consequences. Specifically, the combination between the peculiar symmetry of the altermagnet and the damping by Stoner excitations makes magnons in itinerant altermagnets

completely distinct from their antiferro- and ferromagnetic counterparts. The magnons acquire a strong frequency- and spin-dependent directionality, which can potentially be exploited as a resource in spintronics devices [17].

## 2 Model and mean-field ground state

We model the electronic structure of altermagnets using a Hamiltonian proposed in ref. [18], which is essentially a Hubbard model with an especially chosen hopping structure that realises an altermagnetic symmetry,

$$H = \sum_{ll'} \sum_{\mu\mu'} \sum_{\sigma} \tau_{ll'}^{\mu\mu'} c_{l\mu\sigma}^{\dagger} c_{l'\mu'\sigma} + U \sum_{l,\mu} n_{l\mu\uparrow} n_{l\mu\downarrow}, \tag{1}$$

where $n_{l\mu\sigma} \equiv c_{l\mu\sigma}^{\dagger} c_{l\mu\sigma}$, $l$ and $l'$ label unit cells, $\mu$ and $\mu'$ label sublattices ($A$ or $B$) and $\sigma$ labels the spin projection along the $z$ axis. The hopping matrix $\tau_{ll'}^{\mu\mu'}$ is described in the caption of Fig. 1. The intra-atomic interaction parameter $U$ can be chosen to place the system in either the metallic or insulating altermagnetic phase; for the value of diagonal hopping we adopted in this work, $2\tau \lesssim U \lesssim 3\tau$ yields a metallic altermagnetic phase, whereas $U \gtrsim 3\tau$ produces the insulating altermagnetic phase. The complete mean-field phase diagram of this model has been explored in Ref. [18]. Here we will choose two representative points, one in the insulating and one in the metallic region, and study the elementary spin excitations above their respective mean-field ground states. The mean-field approximation we employ amounts to the following replacement,

$$U \sum_{l,\mu} n_{l\mu\uparrow} n_{l\mu\downarrow} \longrightarrow$$

$$\frac{U}{2} \sum_{l,\mu} \left[ (\bar{n}_{\mu l} + \bar{m}_{\mu l}) n_{l\mu\downarrow} + (\bar{n}_{\mu l} - \bar{m}_{\mu l}) n_{l\mu\uparrow} \right], \tag{2}$$

with $\bar{n}_{\mu l} \equiv \langle n_{l\mu\uparrow} \rangle + \langle n_{l\mu\downarrow} \rangle$ and $\bar{m}_{\mu l} \equiv \langle n_{l\mu\uparrow} \rangle - \langle n_{l\mu\downarrow} \rangle$, plus a constant term that can be safely ignored. The average occupancies $\bar{n}_{\mu l}$ and magnetic moments $\bar{m}_{\mu l}$ are determined self-consistently.

We obtain the magnon spectrum of altermagnets by studying the transverse spin susceptibilities,

$$\chi_{\mu\nu}^{+-}(\vec{r}_{l'} - \vec{r}_l, t) \equiv -i\theta(t) \left\langle \left[ S_{l\mu}^{+}(t), S_{l'\nu}^{-}(0) \right] \right\rangle \tag{3}$$

and

$$\chi_{\mu\nu}^{-+}(\vec{r}_{l'} - \vec{r}_l, t) \equiv -i\theta(t) \left\langle \left[ S_{l\mu}^{-}(t), S_{l'\nu}^{+}(0) \right] \right\rangle, \tag{4}$$

where $t$ is the time, $S_{l\mu}^{-} \equiv c_{l\mu\downarrow}^{\dagger} c_{l\mu\uparrow}$, $(S_{l\mu}^{+} = (S_{l\mu}^{-})^{\dagger})$ is the operator that creates a spin excitation with $S^z = -1$ ($S^z = 1$) at cell $l$ in the sublattice $\mu$, $\vec{r}_l$ is the position of unit cell $l$, and $\theta(t)$ is the Heaviside unit step function. These two-time correlation functions cannot be computed exactly for an interacting model such as the one defined in Eq. 1; the simplest approach that can describe magnons is the so-called random phase approximation (RPA), in which the interaction is taken into account, to all orders in perturbation theory, between the electron and the hole that form the spin-flip excitation [10]. The RPA relates the transverse interacting susceptibilities $\chi^{\perp}$ ($\perp \equiv +-$ or $-+$) to the mean-field susceptibilities $\bar{\chi}^{\perp}$, which are the same Green functions defined in Eqs. 3 and 4, with the thermal average $\langle \cdot \rangle$ evaluated for the mean-field configuration. For the model considered here, after Fourier transforming both in time and position, the RPA equations are

$$\chi_{\mu\nu}^{+-}(\mathcal{Q}) = \bar{\chi}_{\mu\nu}^{+-}(\mathcal{Q}) - U \sum_{\xi} \bar{\chi}_{\mu\xi}^{+-}(\mathcal{Q}) \chi_{\xi\nu}^{+-}(\mathcal{Q}), \tag{5}$$

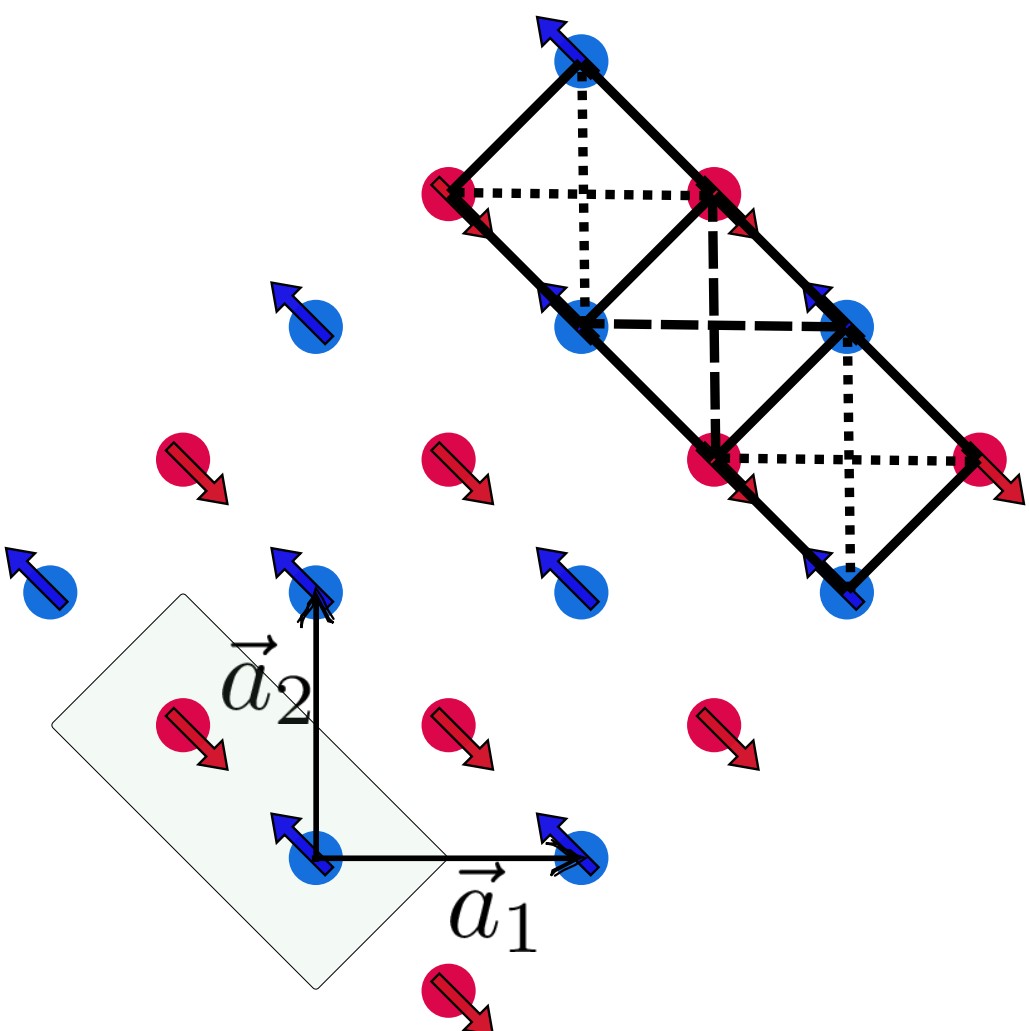

Figure 1: Schematic representation of the model altermagnet on a square lattice defined by primitive vectors $\vec{a}_1$ and $\vec{a}_2$, with $|\vec{a}_1| = |\vec{a}_1| = a$. The solid line connecting blue and red sites represents the nearest neighbor hopping $\tau$. Dashed and dotted lines represent the alternating second neighbor hoppings $\tau'(1 \pm \delta)$. The lightly colored rectangle indicates the unit cells.

90  where $\mathcal{Q} \equiv (\vec{q}, \hbar\Omega)$. We obtain an analogous expression for $\chi^{-+}$. The spectral density associ-
91  ated with magnons, projected on sublattice $\mu$, is given by

$$\rho_\mu^\perp(\mathcal{Q}) = -\frac{1}{\pi}\text{Im}\chi_{\mu\mu}^\perp(\mathcal{Q}) \qquad (6)$$

92  where $\perp$ can be either $+-$ or $-+$, denoting the transversal character of these response functions
93  with respect to the equilibrium staggered magnetization (Néel vector). Magnon energies $\hbar\Omega(\vec{q})$
94  are associated with the positions of the peaks of $\rho^{+-}$ (for the $S^z = -1$ magnons) or $\rho^{-+}$ (for
95  the $S^z = 1$ magnons), at fixed wave-vector $\vec{q}$. Analogously, magnon lifetimes are defined as
96  the inverse of the full width at half-maximum of the magnon peaks.

## 2.1  Mean-field results

98  An insulating altermagnetic state can be obtained by choosing $U \gtrsim 3\tau$; however, for $3\tau \lesssim U \lesssim 10\tau$
99  the mean-field configuration belongs to an intermediate coupling regime, for which the spin
100  dynamics can not yet be properly described by a spin-only (Heisenberg-like) model. Thus, to
101  benchmark our fermionic model against a spin model, we chose $U = 10\tau$, together with the
102  hopping values $\tau' = 0.17\tau$ and $\delta = 0.83$. The self-consistent mean-field solution gives the
103  bands shown in Fig. 6 of appendix A, with a staggered magnetic moment $m_A - m_B = 1.86\mu_B$
104  per unit cell. For the reciprocal space path we plotted in Fig. 6, the spin splitting is zero only
105  along the line $q_y = q_x$. Along the line $q_y = \frac{\pi}{a} - q_x$ there is the characteristic crossing between
106  the $\uparrow$ and $\downarrow$ spin bands, associated with the altermagnetic symmetry.

107      The metallic altermagnetic state can be obtained either by tweaking the hopping param-
108  eters, as shown in ref. [18], or by reducing the Hubbard parameter $U$. We chose the latter
109  option to minimize the differences between the shapes of the electronic bands in the metallic
110  and insulating states. By setting $\tau' = 0.17\tau$, $\delta = 0.83$ and $U = 2.5\tau$ we obtain the metal-
111  lic altermagnetic bands shown in Fig. 6 of appendix A, with a staggered magnetic moment
112  $m_A - m_B = 0.74\mu_B$ per unit cell.

## 3  Magnons

114  To benchmark our methodology, we first analyze the spin excitations of the insulating alter-
115  magnet in the strong coupling limit ($U = 10\tau$), for which the spin model results should be
116  valid [15, 19]. By scanning the spectral densities $\rho^{+-}$ and $\rho^{-+}$ in the $(\hbar\Omega, \vec{q})$ space we obtain
117  the dispersion relations for $S^z = -1$ magnons ($+-$) and for $S^z = 1$ magnons ($-+$), shown in
118  Fig. 2. The energy splitting between the two polarizations, one of the hallmarks of altermag-
119  netism, is clearly seen along high-symmetry directions in the Brillouin zone. We also show
120  the dispersion relation for (linearized) Holstein-Primakoff magnons, extracted from a Heisen-
121  berg model for the altermagnet, including up to third-neighbor exchange. As expected, the
122  agreement with the RPA treatment of the fermionic model is very good in this case [1].

123      Along specific lines within the Brillouin zone we observe a behavior analogous to the "band
124  inversion" associated with topologically non-trivial electronic bands. For instance, along the
125  reciprocal space path going from $(\frac{\pi}{a}, 0)$ to $(0, \frac{\pi}{a})$ there is a crossing between the $S^z = -1$ and
126  the $S^z = 1$ magnon branches. In the presence of spin-orbit coupling a gap may appear at
127  the crossing point $(\frac{\pi}{2a}, \frac{\pi}{2a})$, possibly accompanied by a finite Berry curvature. This crossing is
128  also associated with the peculiar directional behavior of altermagnetic magnons. If we focus
129  on magnons with one $S^z$ value we see that the energy at the $(\frac{\pi}{a}, 0)$ point in reciprocal space
130  (thus, propagating along the $x$ direction in real space with wavelength $\lambda = 2a$) is 40% different

---

[1]This is contrast with the insulating intermediate coupling case, for which the spin model fails. See appendix D.

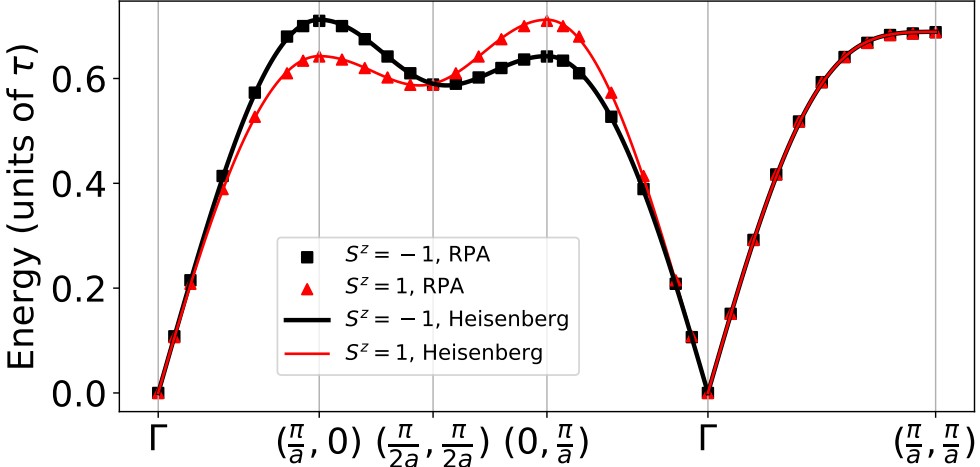

Figure 2: Dispersion relation for magnons in an insulating altermagnet in the strong coupling regime ($U = 10\tau$). The Heisenberg model used to fit the RPA energies includes up to third-neighbor exchange.

from that of a magnon with the same wavelength propagating along the $y$ direction. This is illustrated in fig. 11 of the appendix E, where we plot the magnons spectral densities as a function of propagation direction, for a fixed wavelength. Combined with the fact that, for sufficiently small wavelengths (typically smaller than $\sim 5a$) magnons with a well-defined $S^z$ are strongly sublattice-polarized, this feature may be exploited to guide magnons in spintronics devices.

## 3.1 Itinerant altermagnet

We now turn our attention to the behaviour of magnons in itinerant altermagnets. In contrast to the insulating case, it can be expected that their lifetime is limited by Stoner damping [10, 11, 20]. Magnons with energies exceeding single-particle spin-flip excitations (also known as Stoner excitations), can decay into the Stoner continuum [10]. The magnon lifetime is inversely proportional to the density of Stoner modes, which is given by the imaginary part of the mean-field transverse susceptibility $\bar{\chi}^\perp$. The effect of damping for a conducting altermagnet ($U = 2.5\tau$) is seen in the evolution of the spectral weight of spin excitations, shown along two different directions, $(q, q)$ and $(q, 0)$ with $|q| < \frac{\pi}{a}$, in Figure 3a,b. For low energy, the spectral density has well defined peaks, whose position gives the magnon energy and the inverse of its linewidth gives the magnon lifetime. As the energies are increased, the peaks get broader and, above some energy threshold, they vanish into a continuum. Along the $(q, q)$ direction, both $S_z = \pm 1$ excitations have the same spectral weight (fig. 3a). In contrast, along the $(q, 0)$ direction (fig. 3b), the $S^z = -1$ spin excitations have lorentzian spectral densities with relatively small linewidth in the whole wave number range, whereas the spectral density associated with the $S^z = 1$ spin excitations has a behavior similar to the $(q, q)$ case. We thus find that, for itinerant altermagnets, magnons with a given $S^z$ are only well defined along certain directions.

To make the connection between magnon lifetimes and density of Stoner modes, it is useful to plot both magnons' and Stoner excitations' spectral densities as color-coded functions of energy and wave number, shown in fig. 4. By following the bright spots in the top left panel, it is possible to trace dispersion relations for the $S^z = -1$ magnons, in analogy to the insulating case. For the $S^z = 1$ magnon, the bright spots disappear around $q \sim \frac{2\pi}{5a}$. This can be correlated

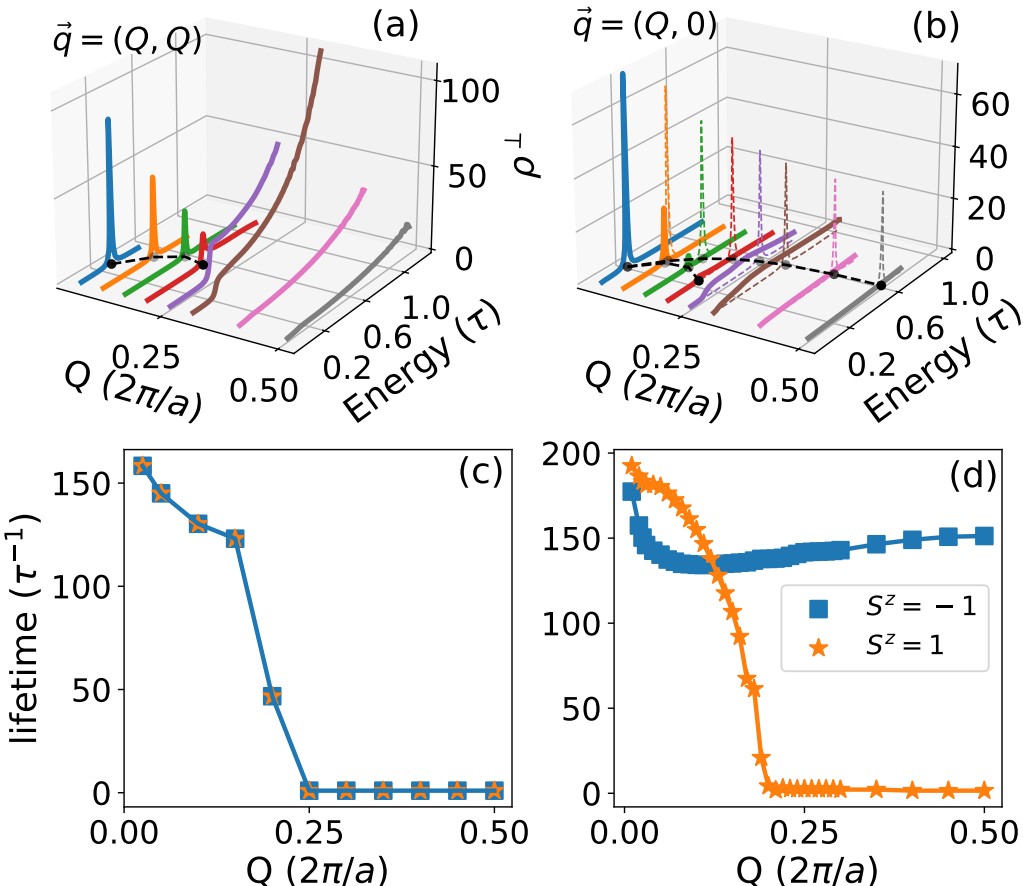

Figure 3: Top: spin excitation spectral densities in the metallic phase ($U = 2.5\tau$), along $\vec{q} = \frac{1}{\sqrt{(2)}}(q,q)$ (a) and $\vec{q} = (q,0)$ (b), as a function of energy, for selected wave numbers. To improve visualization, the spectral density has been multiplied by 100 for the three largest wavenumbers ($q = 0.3$, $0.4$ and $0.5$), by 50 for $q = 0.25$ and by 5 for $q = 0.2$. In (b), solid lines correspond to $\rho^{-+}$, associated with the $S^z = 1$ spin excitations, and dashed lines correspond to $\rho^{+-}$, associated with the $S^z = -1$ spin excitations. Bottom: Lifetimes of the metallic magnons ($U = 2.5\tau$) propagating along the $\vec{q} = \frac{1}{\sqrt{(2)}}(q,q)$ (c) and $\vec{q} = (q,0)$ (d), as a function of wave number, for $S^z = -1$ (squares) and $S^z = 1$ (stars) spin excitations.

160   with the boundaries of the Stoner continuum for $S^z = 1$ spin excitations, plotted in the bottom
161   right panel. In contrast, the density of $S^z = -1$ Stoner modes is uniformly small over the
162   whole wave number and energy ranges where $S^z = -1$ magnons exist. A detailed discussion
163   of the origin of the density of Stoner modes in terms of the geometry of the spin-polarized
164   Fermi surface pockets of the metallic altermagnet is presented in appendix C.

165      The giant magnon-lifetime anisotropy is better seen in a color-coded polar plot of the
166   magnon spectral density, for a fixed wavelength. The angular variable indicates the propa-
167   gation direction, and the radial variable is the magnon energy. In fig. 5 we show such a plot
168   for $\lambda = \frac{10a}{3}$ (wave number $q = \frac{3\pi}{5a}$). The top-left panel shows the spectral density $\rho_A^{+-}$ for
169   $S^z = -1$ magnons, projected on sublattice $A$, and the top-right panel displays the equivalent
170   quantity for sublattice $B$ ($\rho_B^{+-}$). It is clear that $S^z = -1$ magnons are strongly suppressed
171   for angles $\gtrsim 30°$, and the $S^z = 1$ magnons for angles $\lesssim 60°$. Such strong directionality is
172   rarely seen for quasiparticles and elementary excitations, and is potentially very useful for ap-
173   plications, especially when one considers the fact that magnons of wavelengths $\lambda \lesssim 4a$ live

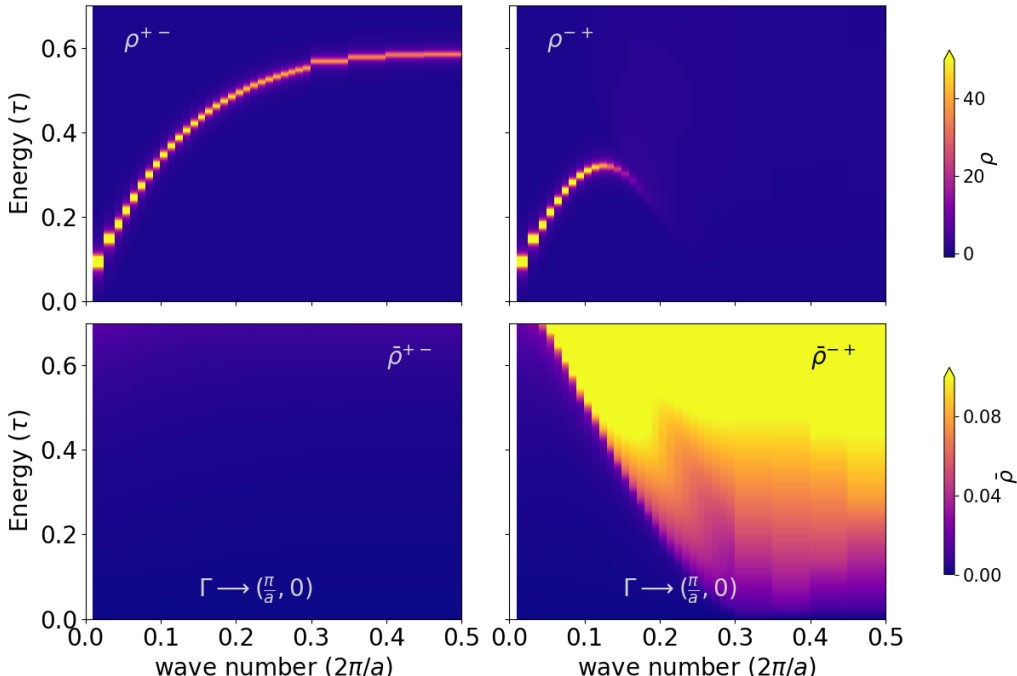

Figure 4: Top: Spectral densities for $S^z = -1$ ($\rho^{+-}$, left) and $S^z = 1$ ($\rho^{-+}$, right) metallic magnons ($U = 2.5\tau$) propagating along the $x$ direction, as a function of wave number and energy. Bottom: Spectral densities for $S^z = -1$ ($\bar{\rho}^{+-}$, left) and $S^z = 1$ ($\bar{\rho}^{-+}$, right) Stoner excitations (single-particle spin flips) propagating along the $x$ direction, as a function of wave number and energy.

preferentially in one of the sublattices. Thus, it is in principle possible to excite magnons along specific directions by choosing their excitation frequency and the sublattice to excite. Selectively addressing the sublattice may be challenging in systems where spin sublattices have atomic size, but not so much in synthetic magnets, where spin sublattices are associated with molecules containing tens of atoms [19, 21].

We have also considered the case of a doped insulating altermagnet, by choosing $U = 3.5\tau$ and imposing an electronic occupation of 1.05 electrons per atomic site. In this case the anisotropic suppression of magnons is observed for propagation angles $30° \lesssim \theta \lesssim 75°$, as shown in the bottom panels of fig. 5. Thus, whenever it is possible to dope an insulating altermagnet electrostatically, it is in principle also possible to control electrostatically the propagation direction of magnons.

The effects of a giant spatial anisotropy in magnon lifetimes are likely to be noticed on several transport coefficients of metallic altermagnets [22]. Electronic transport is expected to be impacted by electron-magnon scattering, especially at low temperatures. Moreover, with current high-resolution spin-polarized electron energy loss spectroscopy [23, 24] it should be possible to probe experimentally the lifetime anisotropy predicted by our theoretical analysis.

We would like to emphasize that the lifetimes of magnons in itinerant magnets is related to the frequency and wave-vector dependent spectral density of Stoner modes, as detailed in the appendix B. The authors of a previous work [9] have estimated the relative intensity of magnon damping, as a function of magnon wave vector only, by integrating the spectral density of Stoner excitations over the whole magnon band width. This quantity can not be associated with the lifetime of individual magnons, although it can give an idea of the overall importance of Stoner excitations for the magnon spectrum. The relevant quantity for determining the lifetime of a magnon with well-defined energy and momentum is the mean-field transverse

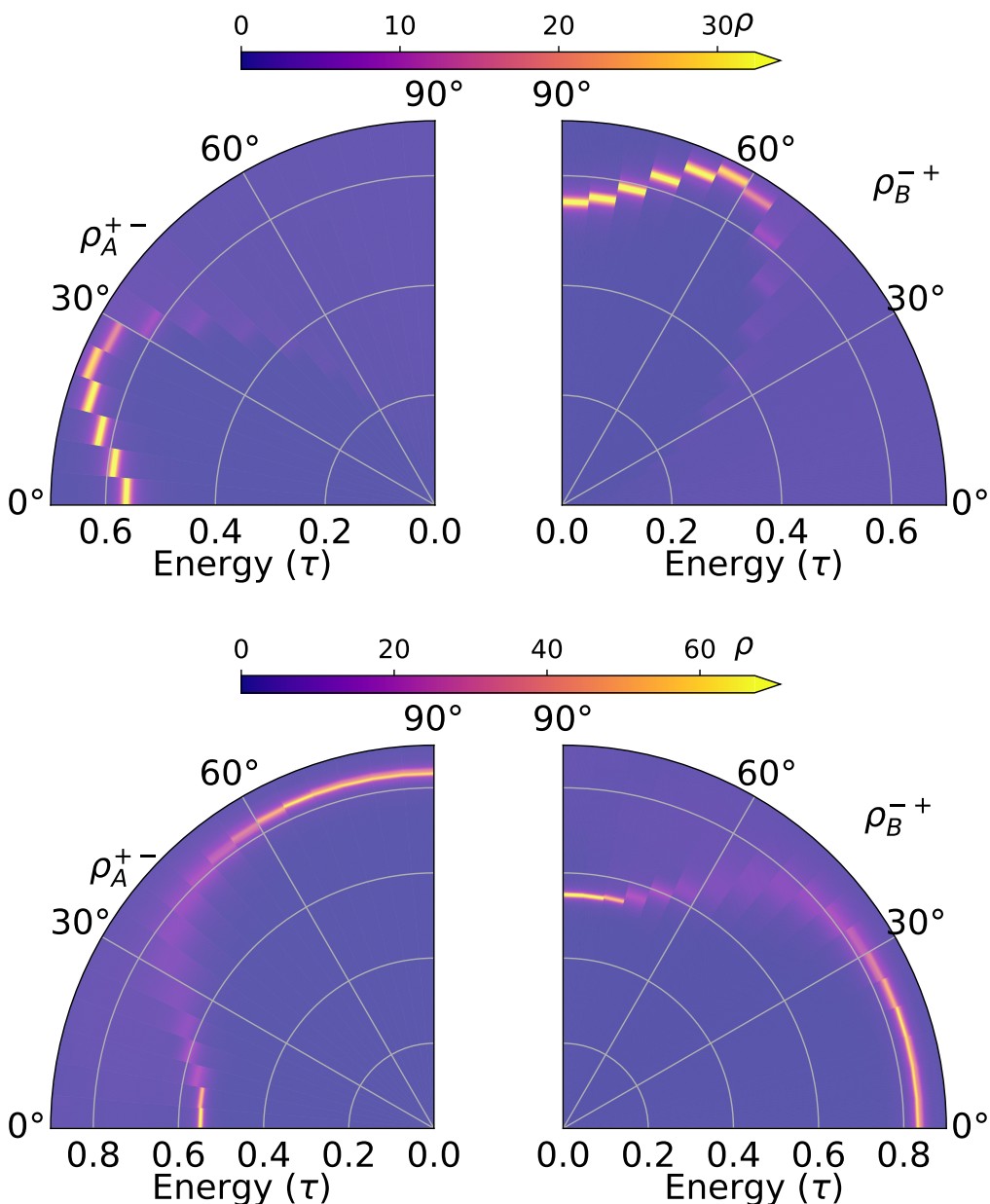

Figure 5: Magnon spectral densities as functions of propagation angle, for a fixed wavelength ($\frac{10a}{3}$). The radial variable represents energy (in units of the nearest-neighbor hopping $\tau$). $\rho_A^{+-}$ corresponds to $S^z = -1$ magnons, $\rho_B^{-+}$ corresponds to $S^z = 1$ magnons. Top panels: metallic phase ($U = 2.5\tau$); bottom panels: doped insulating phase ($U = 3.5\tau$, excess 0.1 electrons per unit cell).

spin susceptibility calculated at the energy of the magnon (the pole of the RPA transverse spin susceptibility), as discussed in appendix B.

# 4 Conclusion

We have studied the intrinsic damping of magnons in altermagnets. These collective modes come with two values of $S_z = \pm 1$. Contrary to their counterparts in ferro- and antiferromagnets, we find a giant spatial anisotropy of magnon lifetimes in itinerant altermagnets. We find that, for a given direction, only magnons with a given sign of $S_z$ survive without melting due to Stoner damping. The ultimate reason for this unique behaviour relies on the existence of spin-polarized Fermi surface pockets that characterizes altermagnets. Therefore, we expect our predictions are generic of all itinerant altermagnets, rather than model specific and will have to be considered in future magnonic applications.

# Acknowledgments

A.T.C. acknowledges fruitful discussions with D. L. R. Santos. The authors acknowledge financial support from FCT (Grant No. PTDC/FIS-MAC/2045/2021), SNF Sinergia (Grant Pimag, CRSII5_205987) the European Union (Grant FUNLAYERS - 101079184). J.F.-R. acknowledges financial funding from Generalitat Valenciana (Prometeo2021/017 and MFA/2022/045), Spanish Government through PID2022-141712NB-C22, and the Advanced Materials programme supported by MCIN with funding from European Union NextGenerationEU (PRTR-C17.I1) and by Generalitat Valenciana (MFA/2022/045).

# A    Mean-field electronic structure

We present the electronic bands corresponding to the mean-field configurations considered in the letter: strong-coupling insulating ($U = 10\tau$, fig. 6, left panel), metallic ($U = 2.5\tau$, fig 6, right panel), and slightly doped insulating ($U = 3.5\tau$, fig. 7, right panel). Both metallic and insulating phases have half-filled bands (one electron per lattice site), whereas the doped phase has 1.05 electrons per lattice site. Table 1 shows the values of the Hamiltonian parameters associated with the different phases, as well as the mean-field staggered magnetic moment per unit cell. We also show the intermediate-coupling insulating case ($U = 3.5\tau$, fig. 7, left panel).

|  | $\tau'$ | $\delta$ | $U$ | $|m_\uparrow - m_\downarrow|(\mu_B)$ |
|---|---|---|---|---|
| Insulating (strong coupling) | 0.16 | 0.83 | 10 | 1.86 |
| Insulating (intermediate coupling) | 0.16 | 0.83 | 3.5 | 1.28 |
| Metallic | 0.16 | 0.83 | 2.5 | 0.74 |

Table 1: Values for the Hamiltonian parameters (in units of the nearest-neighbor hopping $\tau$) used in this work, and respective staggered magnetic moment per unit cell, in units of Bohr magnetons $\mu_B$.

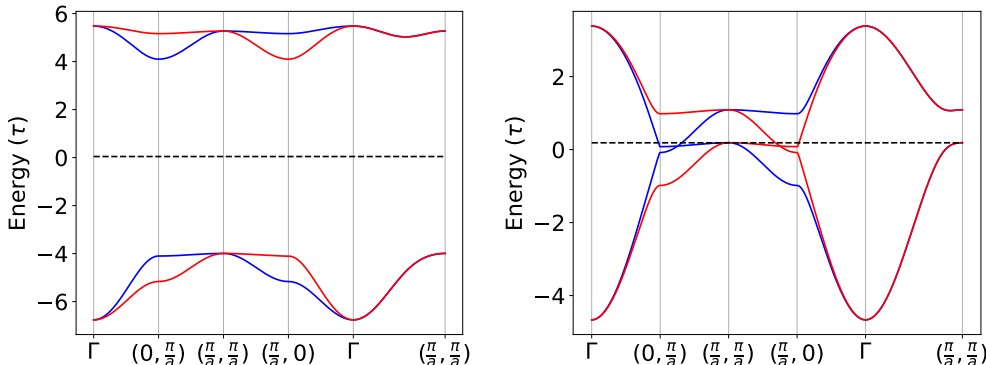

Figure 6: Electron energy bands for the strong-coupling insulating (left panel, $U = 10\tau$) and metallic (right panel, $U = 2.5\tau$) mean-field ground state configuration of the altermagnet Hamiltonian (eq. 1 of the main text), with $\tau' = 0.16\tau$ and $\delta = 0.83$. Red and blue lines represent $\uparrow$ and $\downarrow$ spin sub-bands. The black dashed line marks the Fermi energy.

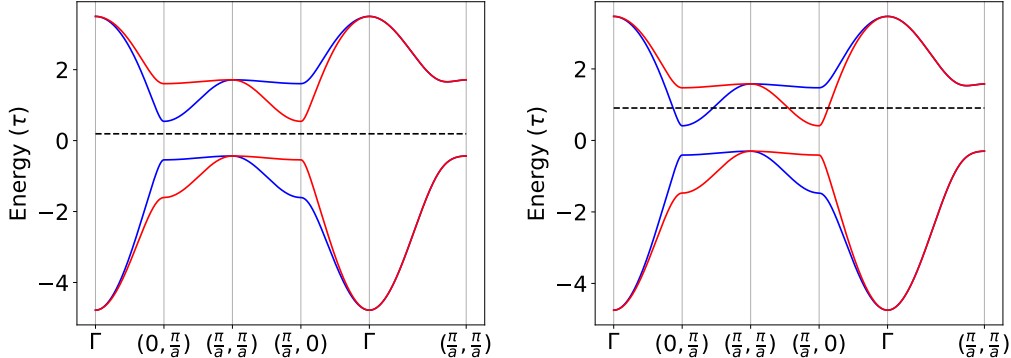

Figure 7: Electron energy bands for the mean-field ground state configuration of the altermagnet Hamiltonian (eq. 1 of the main text) in the insulating intermediate coupling regime ($U = 3.5\tau$) at half-filling (left panel) and away from half-filling (1.05 electrons per lattice site, right panel). The values for the hopping parameters are $\tau' = 0.16\tau$, $\delta = 0.83$. Red and blue lines represent $\uparrow$ and $\downarrow$ spin sub-bands. The black dashed line marks the Fermi energy.

## B  Relationship between the density of Stoner modes and the magnon lifetime

The standard random phase approximation (RPA) applied to the transverse spin susceptibility of a Hubbard Hamiltonian results in a relationship between the magnon Green function $\chi^{+-}$ and the mean-field Green function $\bar{\chi}^{+-}$,

$$\chi^{+-}(\vec{q}, \hbar\Omega) = \frac{\bar{\chi}^{+-}(\vec{q}, \hbar\Omega)}{1 + U\bar{\chi}^{+-}(\vec{q}, \hbar\Omega)}. \tag{B.1}$$

We would like to cast this expression in a form that resembles a Green function with a self-energy correction,

$$G = \frac{1}{\bar{G}^{-1} + \Sigma}, \tag{B.2}$$

where $\bar{G}$ is the bare Green function and $\Sigma$ is the self-energy. For this it is useful to split all quantities into their real and imaginary parts, denoted below by $R$ and $I$ subscripts. The real and imaginary parts of the magnon Green function then become (we will omit the energy and wave vector arguments for now to avoid cluttering the expressions)

$$\mathrm{Re}\left[\chi^{+-}\right] = \frac{\bar{\chi}_R^{+-}(1 + U\bar{\chi}_R^{+-}) + U(\bar{\chi}_I^{+-})^2}{(1 + U\bar{\chi}_R^{+-})^2 + (U\bar{\chi}_I^{+-})^2},$$

$$\mathrm{Im}\left[\chi^{+-}\right] = \frac{\bar{\chi}_I^{+-}}{(1 + U\bar{\chi}_R^{+-})^2 + (U\bar{\chi}_I^{+-})^2}. \tag{B.3}$$

Similarly,

$$\mathrm{Re}\left[G\right] = \frac{\bar{G}^{-1} + \Sigma_R}{(\bar{G}^{-1} + \Sigma_R)^2 + \Sigma_I^2},$$

$$\mathrm{Im}\left[G\right] = -\frac{\Sigma_I}{(\bar{G}^{-1} + \Sigma_R)^2 + \Sigma_I^2}. \tag{B.4}$$

By comparing the imaginary parts of the generic Green function $G$ to $\mathrm{Im}\left[\chi^{+-}\right]$ we notice immediately a clear analogy between $U\bar{\chi}_I^{+-}$ and $\Sigma_I$. Notice also that, as in the electronic case, magnon damping is inextricably tied to shifts in magnon energy, through the real part of the self-energy $\Sigma_R$. It is clear, then, that the lifetime of a magnon with wave vector $\vec{q}$ and energy $\hbar\Omega(\vec{q})$ is determinmed by the spectral density of Stoner modes with wave vector $\vec{q}$ and energy $\hbar\Omega(\vec{q})$.

## C  Origin of the anisotropic magnon lifetime

To further shed light on the mechanism behind the lifetime anisotropy of metallic magnons, it is useful to look at constant energy contours of the electronic bands in the mean-field al-termagnetic configuration. The goal is to identify qualitatively the direction dependence of single-particle spin-flip transitions that give rise to the anisotropic density of Stoner modes. In figure 8 we show three constant energy contours for each spin direction, blue contours for ↑ spin electrons, red contours for ↓. In the left panel we show contours for occupied ↑ states (including the Fermi contour at zero energy) and unoccupied ↓ states (also including the Fermi contour at zero energy), relevant for $S^z = -1$ spin flips (↓⟶↑). Thus, in the left panel we can identify possible single-particle spin-flip transitions by connecting blue and red contours.

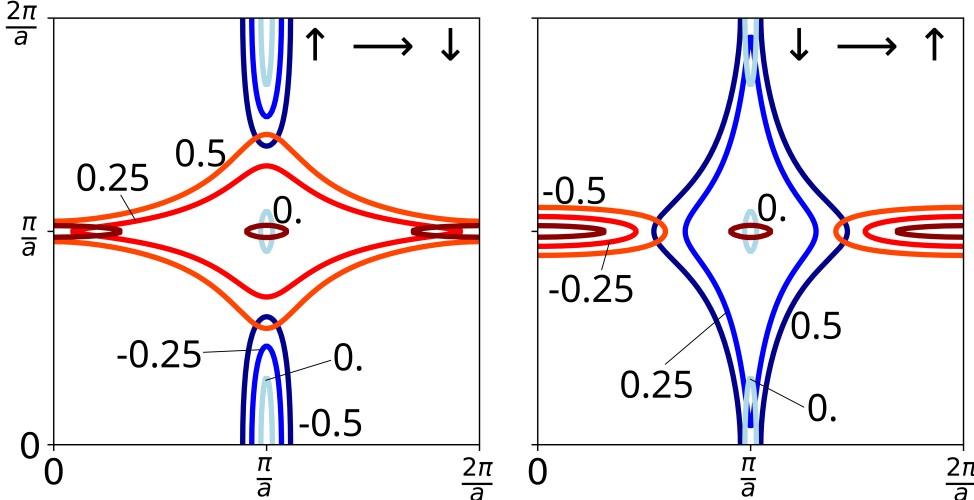

Figure 8: Contours of the electronic bands around the Fermi level; blue curves are for ↑ spin bands, red curves for ↓. Left panel (↑⟶↓): occupied ↑ states (shades of blue, at energies $E_F - 0.5\tau$, $E_F - 0.25\tau$ and $E_F$) and unoccupied ↓ states (shades of red, $E_F$, $E_F + 0.25\tau$ and $E_F + 0.5\tau$). Right panel (↓⟶↑): occupied ↓ states (shades of red, at energies $E_F - 0.5\tau$, $E_F - 0.25\tau$ and $E_F$) and unoccupied ↑ states (shades of blue, $E_F$, $E_F + 0.25\tau$ and $E_F + 0.5\tau$).

In the left panel we see that, apart from the very small pockets at $(\frac{\pi}{a}, \frac{\pi}{a})$, there is no horizontal line connecting blue and red contours. The consequence is that the density of $S^z = -1$ Stoner modes with wave vectors along the $x$ direction is very small, and $S^z = -1$ magnons propagating along the $x$ direction are long-lived. On the other hand, there are plenty of connections between blue and red contours at angles $\gtrsim 30°$, meaning that magnons propagating along those directions will be substantially damped. In the right panel we show the analogous information for $S^z = 1$ spin flips (↓⟶↑): occupied ↓ states (including the Fermi contour at zero energy) and unoccupied ↑ states (also including the Fermi contour at zero energy). Now it is clear that there are many possible single- particle spin-flip transitions with wave vectors along $x$, whereas very few with wave vectors along $y$, thus meaning that $S^z = 1$ magnons are strongly damped when propagating along $x$ but long-lived when propagating along $x$.

## D   Insulating alternagnet in the intermediate coupling regime ($U = 3.5\tau$).

As mentioned in the main main text, the insulating alternagnetic phase of the model is obtained for $U \gtrsim 3\tau$. In this regime, although the electronic bands are clearly those of an alternagnetic insulator (see the left panel of figure 7), the magnons bear marks of itinerant magnetism, especially at short wavelengths. A clear signature of itinerant behavior is the fact that the magnon lineshape acquires a finite linewidth and, at large enough energies, deviates significantly from a a lorentzian shape. This is seen in fig. 10 for a short wavelength magnon ($\lambda = 2a$) propagating along the $x$ direction. The lineshape of the $S^z = 1$ magnon (right panel) is very close to a lorentzian (dashed orange line). In contrast, the lineshape of the higher energy $S^z = -1$ magnon (left panel) is clearly not a lorentzian.

Another consequence of the coupling between magnons and Stoner excitations is a renormalization of magnon energies relative to those predicted by a localized spin model. In fig. 9 we compare the dispersion relation of magnons for the insulating alternagnet in the intermediate coupling regime, extracted from the fermionica model, to the energies of linearized

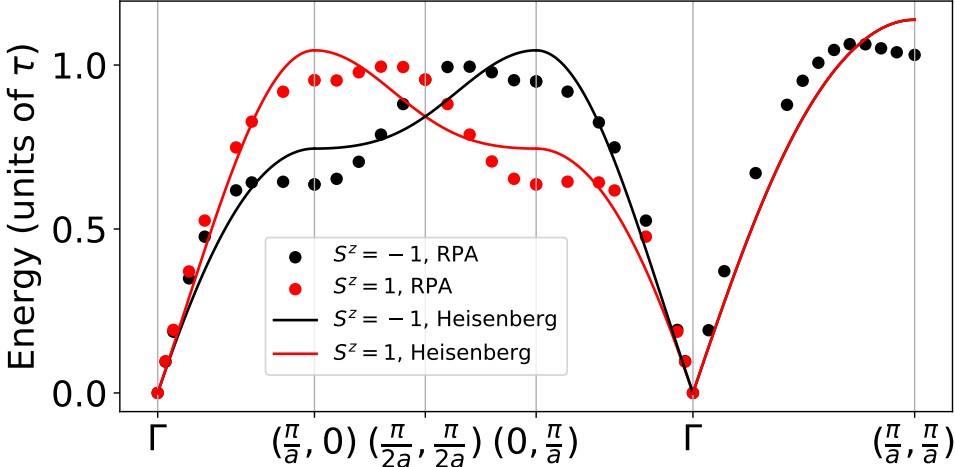

Figure 9: Dispersion relation for magnons in an insulating altermagnet in the intermediate coupling regime ($U = 3.5\tau$). The Heisenberg model used to fit the RPA energies includes up to third-neighbor exchange.

Holstein-Primakoff magnons of a localized spins model, with exchanges up to third neighbors. The exchange parameters of the localized spin model have been obtained from a fit to the fermionic model energies. Although the main qualitative features of the dispersion are captured by the localized spins model, it does a poor job of matching quantitatively the magnon energies over the whole Brillouin zone, since the spin only model cannot capture the renormalization of the magnon energies by Stoner excitations.

To illustrate the effect of the coupling to the Stoner continuum we plot, in fig. 10, the spectral densities for magnons with $S^z = -1$ ($\rho^{+-}$) and $S^z = 1$ ($\rho^{-+}$). Notice that the lineshape of the $S^z = -1$ magnon (left panel) is clearly not a lorentzian, whereas the $S^z = 1$ magnon is well fitted by a lorentzian with a finite linewidth, denoting a finite lifetime.

## E  Directionality of the magnon spectrum in the insulating regime (intermediate coupling).

Here we illustrate the directional dependence of the magnon energies for the intermediate coupling ($U = 3.5\tau$) insulating case (figure 11). The main difference between this case and the metallic and slightly doped cases is that the magnons appear as well-defined collective excitation for all directions of propagations (compare with fig. 5 of the main text).

A very good agreement between the predictions of the fermionic model and those of the spin-only model is expected for insulating magnets in general; here the less-than-perfect agreement can be partially attributed to the influence of Stoner excitations for wave vectors close to the edges of the Brillouin zone. Fig. XXX in the SM shows that the spectral density for high-energy, short wavelength magnons are significantly broadened, a consequence of damping by Stoner excitations. [11, 20] Another possible reason for the disagreem

This is due to the continuum of Stoner modes with vanishingly small energies and wave numbers, coming from states around the two Fermi surface pockets with opposite spins centered at $(\frac{\pi}{a}, \frac{\pi}{a})$ (see Fig. 8 of appendix C). As the magnon energy increases (and wavelength decreases), the lifetime of one of the polarizations has a slight monotonic decrease over the whole energy range, while for the opposite polarization it has similar behaviour up to $\lambda \lesssim 4a$,

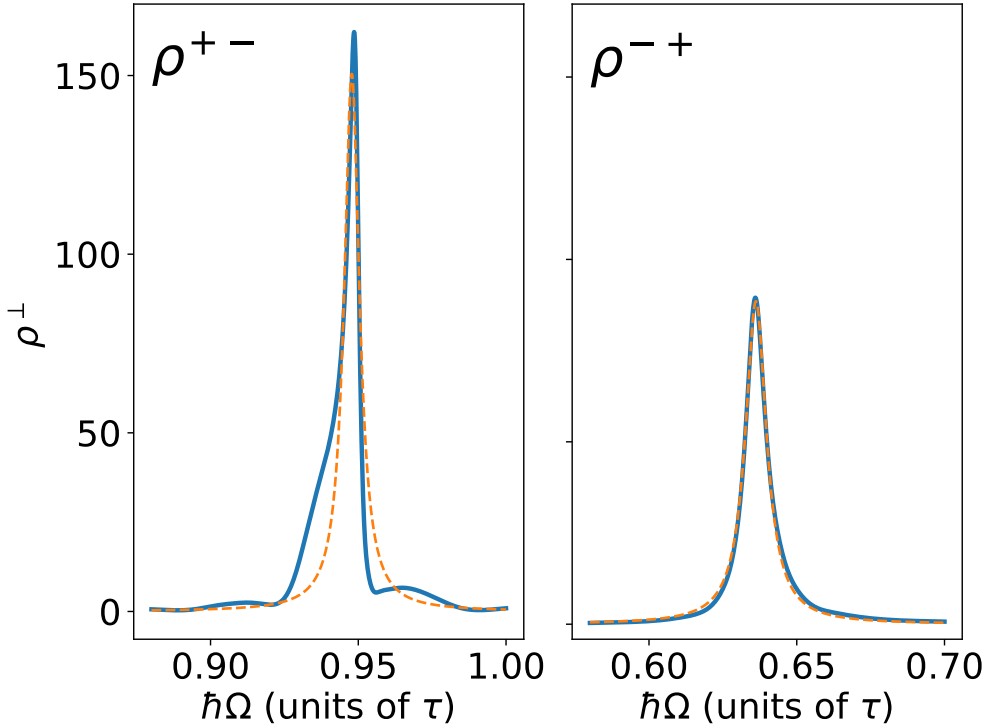

Figure 10: Spectral density of insulating magnons in the intermediate regime ($U = 3.5\tau$), for wave vector $\vec{q} = (\frac{\pi}{a}, 0)$. The left and right panels correspond to $S^z = -1$ and $S^z = 1$ magnons, respectively.

where the lineshape starts to change drastically, leading to the disappearance of the magnon signal altogether. This is shown in fig. 4 for propagation along the $x$ direction. In fact, the shape of $\rho^{-+}$ deviates so much from the Lorentzian shape (associated with long-lived quasi-particles) that the $S^z = 1$ magnon essentially vanishes. This time the culprit is the Stoner continuum associated with single-particle excitations from the $\downarrow$ states around $(\frac{\pi}{a}, 0)$ to the $\uparrow$ states around $(\frac{\pi}{a}, \frac{\pi}{a})$, whose spectral density is shown in the right bottom panel of fig. 4). The large spectral density for these single-particle spin flip excitations occur for wave vectors predominantly pointing along the $x$ direction, thus producing highly anisotropic, polarization dependent damping rates. In contrast, the spectral density for single-particle excitations with $S^z = -1$ propagating along $x$, shown in the left-bottom panel of fig. 4, is very small over the whole $(q_x, \hbar\Omega)$ plane. This ensures the $S^z = -1$ magnons propagating along $x$ are well-defined collective excitations over their whole bandwith. For magnons propagating along $y$ hte picture is reversed, with $S^z = -1$ magnons strongly suppressed for wavelengths $\lambda \lesssim 5a$.

For wave vectors along the line $q_x = q_y$, where magnons of both polarizations are degenerate, the spectral densities for both $S^z = \pm1$ magnons become featureless for $\lambda \lesssim 4a$, indicating that magnons along that direction are completely suppressed by damping. This is shown in fig. 3a, where we plot the spectral density as a function of energy for selected wave numbers. Once again, this suppression can be understood as a result of critical damping by the Stoner continua connecting states around the Fermi surface pockets centered at $(0, \frac{\pi}{a})$ and $(\frac{\pi}{a}, 0)$ to the states around the Fermi pocket centered at $(\frac{\pi}{a}), \frac{\pi}{a})$. In this case, though, both single-particle spin flips are allowed, $\uparrow \longrightarrow \downarrow$ and $\downarrow \longrightarrow \uparrow$.

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

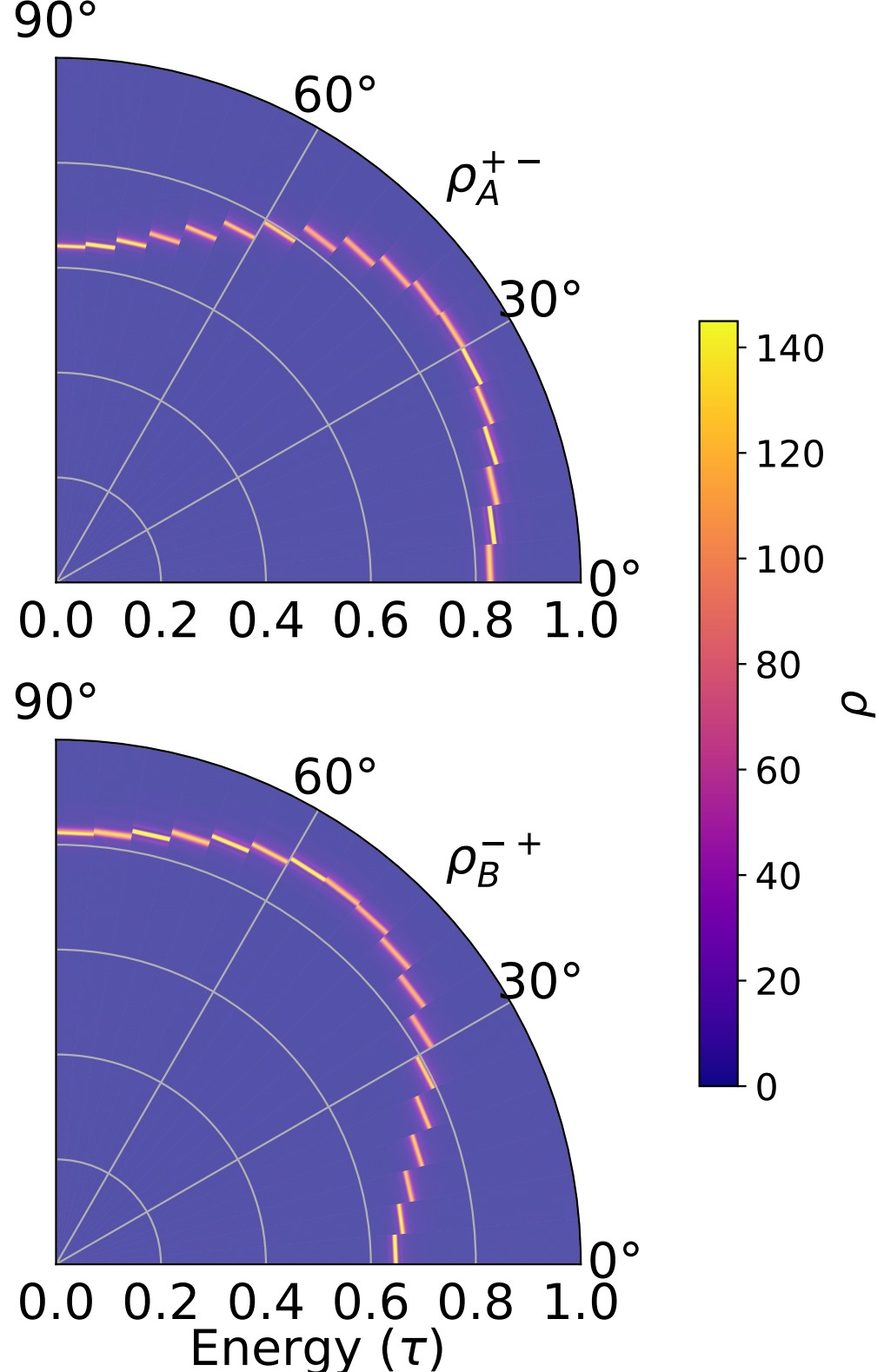

Figure 11: **Directionality of magnons in an insulator**. We plot the magnon spectral densities, as a function propagation angle, for a fixed wavelength ($\frac{10a}{3}$) for an insulating altermagnet. The top panel shows $\rho_A^{+-}$ (for the $S^z = -1$ polarization) and the bottom panel shows $\rho_B^{-+}$ (for the $S^z = -1$ polarization). The radial variable represents energy (in units of the nearest-neighbor hopping $\tau$).