# Peer review of "Giant spatial anisotropy of magnon Landau damping in altermagnets"

_SciPost Physics, doi:SciPost Phys. 18, 125 (2025)_

## Round 1 · Referee Report · Anonymous (Referee 1) · 2024-12-27

Report

The authors of the study “Giant spatial anisotropy of magnon lifetime in altermagnets”, have investigated magnons in altermagnets and many-body effects on them with special focus on stoner excitations. For this purpose, they have adopted a Hubbard model of altermagnetism that maintains the parametric continuous transition from the metallic phase to the insulating phase altermagnets and includes the important features of altermagnets such as non-relativistic spin splitting band structure as well as the correlation effects. According to their findings, altermagnetism which comes as a result of reducing the symmetry of the lattice gives rise to interesting spatial anisotropy of magnons with unique directionality of the magnon propagation and lifetime in metallic altermagnets. They relate their finding in directionality dependence of magnon excitation and propagation to broken chirality of the two magnon dispersion which is a unique feature of altermagnets. In addition, they show that even when magnons could be excited in a certain direction with well-defined quasiparticle character, they might be highly suppressed at higher energies and wavevectors due to the Stoner excitations. Nonetheless, their findings reveal that magnons in insulating altermagnets agrees very well with linear spin wave theory and therefore the correlation effects are negligible in this type of altermagnets.

I found the study very intriguing and interesting in many aspects such as its novelty, well-written manuscript, and advanced theory. I highly recommend this manuscript for the publication in SciPost Physics. Nonetheless, there are minor issues that I would like to ask the authors to address.
1. After the Hamiltonian in equation (1) is introduced, the parameters are not well described which makes it difficult for the reader to understand. I suggest a complete description of the model and all its parameters to be given after equation (1).
2. It is mentioned that spin dynamics in the intermediate regime (3τ<U<10τ) the spin dynamics cannot be properly described by a spin-only model. The authors should provide some more evidence and if possible or provide some references for the readers.
3. In the case of the insulating altermagnets, it is said that magnon energy of one channel is 40% different from the magnon energy of the other channel. However, in Figure 2, the difference does not look like to be that large. Would you please clarify this confusion?
4. Appendix E is very disorganized. There is no logical connection between paragraphs. I suggest revisiting it.

Recommendation

Ask for minor revision

  • validity: -
  • significance: -
  • originality: -
  • clarity: -
  • formatting: -
  • grammar: -

Author:  António Costa  on 2025-02-11  [id 5210]

(in reply to Report 1 on 2024-12-27)
Category:
answer to question

1.
We thank the referee for the suggestion. An improved description of the structure of the hopping matrix has been added to the text just after Eq. 1. We also added legends to fig, 1 to make the meaning of the various elements clearer.

2.
We have provided two pieces of evidence that we believe should convince the readers of the limitations of the spin-only model in the intermediate coupling regime . The first one is the poor fitting between the spin-only dispersion relation and the magnon energies extracted from the fermionic model (Fig. 9). The second, more conclusive one is the clear departure from the simple pole structure in the magnon spectral density, expected for magnons well described by spin-only models. This is shown in Fig. 10. We have added a reference in which spin models fail to describe magnons in a different context (new ref. 19).

3.
We thank the referee for pointing that out. We have mixed up the discussion of the strong coupling limit (which appears in the main text) and the intermediate coupling limit (which is presented in the appendix). The referee is absolutely right that the energy differences in the strong coupling limit are considerably smaller than 40%. The larger differences appear in the intermediate coupling limit. We have amended the text around Fig. 2 accordingly, and transferred the statement about the intermediate coupling regime to appendix D.

4.
We thank the referee for pointing that out. Appendix E was indeed very disorganized. We have organized it in the revised version.

---

## Round 1 · Referee Report · Anonymous (Referee 2) · 2025-1-7

Strengths

The manuscript is timely as there is a lot of interest in electronic and spin properties of altermagnets in general, with recently opened (by Ref. [9]) exploration of magnon excitation in this novel class of magnetic materials. The findings of the paper are novel and interesting, as the theory presented in Ref. [9] is simplistic one assuming noninteracting quasiparticles, which seems to be insufficient according to present papers and some other studies completed after Ref. [9].

Weaknesses

No major weakness, but some improvement in presentation is needed.

Report

The paper presents perhaps one of the first exploration of broadening of magnon bands from simplistic linear spin wave theory (used in Ref. [9]) due to interaction with other excitations in magnetic solids, such as Stoner excitations in the case of the present manuscript. This topic has been explored for conventional metallic ferromagnets [PHYSICAL REVIEW B 84, 174418 (2011)], as well as in Ref. [9], but conclusions of Ref. [9] are quite different from the present manuscript. Due to impact of spatial anisotropic magnon lifetime on future experimental studies and technological applications of altermagnetic magnons.

Requested changes

  1. The authors label the effect they study "Stoner damping", but it would be better aligned with the rest of the literature to call it "Landau damping" (which is due to Stoner excitations). This is terminology used in Ref. [9], prior papers exploring it for conventional magnets, see, e.g., PHYSICAL REVIEW B 84, 174418 (2011) on "Different dimensionality trends in the Landau damping of magnons in iron, cobalt, and nickel: Time-dependent density functional study".

  2. The title of the paper does not really highlight the main effect studied as there are many potential reasons for finite magnon lifetime. Akin to reference mentioned in 1., it would be better to add "magnon lifetime due to Landau damping" into the title.

  3. The authors of Ref. [9] have apparently done extensive study of Landau damping effect on altermagnetic magnons, concluding that: "We also show that, overall, the Landau damping of this metallic altermagnet is suppressed due to the spin-split electronic structure, as compared to an artificial antiferromagnetic phase of the same RuO2 crystal with spin-degenerate electronic bands and chirality-degenerate magnon bands." As this is strikingly different from the conclusion of the present manuscript, so some discussion the origin of this discrepancy is necessary.

  4. Eq. 1 would be far easier to understand if all of its labels were also placed in Fig. 1, meaning add label to each site, add labels on dashed lines which represent hopping etc.

  5. Another Hubbard model for altermagnets was proposed in https://doi.org/10.1103/PhysRevB.108.L100402. It would be good to mention it and/or compare to Eq. 1.

Recommendation

Ask for minor revision

  • validity: high
  • significance: high
  • originality: good
  • clarity: good
  • formatting: reasonable
  • grammar: excellent

Author:  António Costa  on 2025-02-11  [id 5211]

(in reply to Report 3 on 2025-01-07)
Category:
remark
answer to question
reply to objection
correction

  1. We agree with the referee that the most common term used in the literature is Landau damping. We have replaced all occurrences of “Stoner damping” by “Landau damping.”

2. We have changed the title to address the referee's concern. We have also included a sentence in the abstract to stress the fact that the lifetimes we talk about in this manuscript are caused by Landau damping by Stoner excitations.

3. We have indeed discussed the results of ref. 9 in contrast to ours. The discussion appears in lines 190 to 199 of the originally submitted manuscript.

4. We thank the referee for the suggestion. We have added appropriate labels to fig. 1 to make it easier to interpret and more informative.

5. The model proposed in the work mentioned by the referee is essentially the same as the one we employed, with slightly different notation. We have cited this work in our manuscript (Ref. 5).

---

## Round 2 · Referee Report · Anonymous (Referee 1) · 2025-3-1

Strengths

1- Advance model that could capture most of the microscopic properties of real correlated materials. 2- Accurate description of the microscopic origin of the observed magnon properties of altermagnets. 3- Consideration of a wide range of scenarios such as metallic and insulating phases of altermagnets.

Report

The authors of the study “Giant spatial anisotropy of magnon lifetime in altermagnets”, have investigated magnons in altermagnets and many-body effects on them with special focus on stoner excitations. For this purpose, they have adopted a Hubbard model of altermagnetism that maintains the parametric continuous transition from the metallic phase to the insulating phase altermagnets and includes the important features of altermagnets such as non-relativistic spin splitting band structure as well as the correlation effects. According to their findings, altermagnetism which comes as a result of reducing the symmetry of the lattice gives rise to interesting spatial anisotropy of magnons with unique directionality of the magnon propagation and lifetime in metallic altermagnets. They relate their finding in directionality dependence of magnon excitation and propagation to broken chirality of the two magnon dispersion which is a unique feature of altermagnets. In addition, they show that even when magnons could be excited in a certain direction with well-defined quasiparticle character, they might be highly suppressed at higher energies and wavevectors due to the Stoner excitations. Nonetheless, their findings reveal that magnons in insulating altermagnets agrees very well with linear spin wave theory and therefore the correlation effects are negligible in this type of altermagnets.

Recommendation

Publish (easily meets expectations and criteria for this Journal; among top 50%)

---

## Round 2 · Referee Report · Anonymous (Referee 2) · 2025-3-6

Report

This is an important work in the rapidly developing field of altermagnets. It is also quite sophisticated, using a quantum many-body approach required for the complexity of these materials, when compared to the flood of papers using single-particle quantum mechanics formalisms. The authors have improved the manuscript and its explanations/terminology, so it can be now published as is. I would only suggest checking, during proof correction process, all cited arXiv papers and update those have been published since original submission.

Recommendation

Publish (surpasses expectations and criteria for this Journal; among top 10%)

---

## Round 2 · Author Response

Dear Editor,

We hereby submit a revised version of our manuscript "Giant spatial anisotropy of magnon Landau damping in altermagnets" for your appreciation. There was a slight change in title to comply with one of the Referee's requests. Other than that, all changes were minor, but certainly improving the manuscript clarity and readability.

Sincerely yours,
António Costa
(for the authors)

---

## Round 2 · List of Changes

• Title changed to "Giant spatial anisotropy of magnon Landau damping in altermagnets," indicating explicitly the origin of magnons' finit lifetimes considered in the manuscript.

  • The phrase "due to Landau damping caused by coupling to Stoner modes" has been added to the abstract.

  • Fig. 1 has new labels to facilitate identification of the different hopping terms (in the previous version the meaning of the various symbols were indicated in the figure caption only).

  • "Stoner damping" has been replaced by "Landau damping by Stoner modes" at line 54.

  • A more detailed description of the hopping terms in the Hamiltonian has been added after Eq. 1, at lines 65-69.

  • A new bibliography reference ([19]) has been added to footnote 1 at page 5.

  • A discussion of the energy difference between different magnon flavors has been amended (lines 134-138).

  • "Stoner damping" has been replaced by "Landau damping by Stoner modes" at lines 144-145 and at line 210.

  • Appendix E has been reformulated for organization and clarity.

---

## Editorial Decision

published